# Impact of Physical Activity on COVID-19

**DOI:** 10.3390/ijerph192114108

**Published:** 2022-10-28

**Authors:** Jia Yang, Xiang Li, Taiyu He, Fangyuan Ju, Ye Qiu, Zuguo Tian

**Affiliations:** 1College of Physical Education, Hunan University, Changsha 410012, China; 2Chongqing Medical University, Chongqing 400016, China; 3College of Physical Education, Yangzhou University, Yangzhou 225012, China; 4College of Biology, Hunan University, Changsha 410012, China

**Keywords:** SARS-CoV-2, COVID-19, physical activity

## Abstract

Introduction: The coronavirus disease 2019 (COVID-19) pandemic, which is caused by severe acute respiratory syndrome coronavirus 2 (SARS-CoV-2), is seriously endangering human health worldwide. This study finds effective intervention modalities of physical activity on COVID-19 through a narrative review. Methods: In this study, 41 papers were selected for a narrative literature review after a comprehensive database search from 20 December 2019, to 30 August 2022. Results: 41 articles meet the established criteria, and in this review, we comprehensively describe recent studies on exercise and COVID-19, including the impact and recommendations of exercise on COVID-19 prevention, patients with COVID-19, and noninfected populations. Conclusions: The literature suggests that physical activity (PA) contributes to the prevention and treatment of COVID-19, can promote recovery of physical function, alleviate post-acute COVID-19 syndrome, and improve patients’ psychological well-being. It is recommended to develop appropriate exercise prescriptions for different populations under the guidance of a physician.

## 1. Introduction

The COVID-19 pandemic caused by severe acute respiratory syndrome coronavirus 2 (SARS-CoV-2) is still prevalent worldwide. As of 14 August 2022, World Health Organization (WHO) has reported 587 million confirmed COVID-19 cases, including 6.4 million deaths [1]. COVID-19 patients can be asymptomatic or symptomatic. Symptomatic patients may develop fever, dyspnea, fatigue, muscle pain, headache, new loss of taste or smell, sore throat, runny nose, nausea, diarrhea, and even death in severe cases [2,3]. Meanwhile, COVID-19 has different effects on patients with different characteristics. For instance, symptoms such as fever, systemic inflammatory response syndrome, respiratory insufficiency, bilateral pneumonia, acute cardiac injury, and renal failure were significantly more frequent in male patients, while vomiting, diarrhea, and hyposmia/anosmia were more common in female patients [4,5]. Furthermore, among patients hospitalized with COVID-19, men tended to have higher mortality than women [4,5]. This disparity in the prognosis of COVID-19 between men and women may be caused by differences in sex hormones, sex chromosomes, underlying diseases, and gender-specific behaviors [6,7,8]. Even after recovery from acute COVID-19, some patients may still have long-term post-acute COVID-19 syndromes, such as persistent dyspnea, decreased physical function, and hypoxia, resulting in a decline in life quality [9].

Numerous studies confirmed that physical activity (PA) could provide physical and psychological benefits and contribute to the prevention and treatment of various diseases, such as cardiovascular disease, diabetes, cancer, hypertension, obesity, depression, and osteoporosis [10,11,12]. There are four types of PA, including aerobic activity, muscle-strengthening activity, bone-strengthening activity, and multicomponent physical activity, with different benefits [12]. Children, adolescents, adults, older adults, pregnant/postpartum women, and patients with chronic conditions or disabilities can refer to the guidelines for appropriate exercise to improve their health [12,13]. Recent studies have reported that PA may help prevent and treat COVID-19, improve patients’ mental health, promote the recovery of physical functions, and alleviate post-acute COVID-19 syndrome [14,15,16,17]. Therefore, to better understand the impact of PA on COVID-19, we conducted a review of the studies related to exercise and COVID-19. This review may provide a basis for developing safe and effective exercise strategies for healthy people and COVID-19 patients.

## 2. Materials and Methods

We performed a narrative review of the role and effect of physical activity with the COVID-19 coronavirus; the study was carried out according to the methodology proposed by Gasparyan et al. [18].

### 2.1. Search Strategy

A comprehensive search of the following databases MEDLINE, EMBASE, Scopus, PubMed, and Web of Science was conducted until 30 August 2022. These studies investigate the effects of exercise interventions on novel coronaviruses. Key terms, including medical subject headings (MESH) terms. The MESH terms used were “COVID-19”, “SARS-CoV-2”, “COVID-19 syndrome”, acute respiratory distress syndrome (ARDS), “physical activity (PA)”, “Sports”, “physical exercise”, “post-COVID syndrome (PCS)”, “early ambulation”, “physical therapy modalities”, “rehabilitation”, and “telerehabilitation”. All key terms were searched in titles and abstracts. All key terms were written in English, and when the database had the idiom filter tool, the language was restricted to English, Portuguese and Spanish. This review manually checked the content and information in the articles for compliance with the scope of this study to identify eligible literature more accurately. The search was conducted independently by two authors, who identified and removed duplicates. The target literature was first targeted by title and abstract, and then the full text was reviewed to consider eligibility inclusion and exclusion criteria. Disagreements were resolved by consensus with the third author.

### 2.2. Eligibility Criteria

Inclusion criteria were articles that examined the impact of physical activity on COVID-19 prevention, treatment, and recovery. Exclusion criteria are for Complications of COVID-19, special populations, and articles on the effects of COVID-19-specific drugs on exercise (Figure 1).

## 3. Results

A total of 671 studies were identified in the search, and after excluding 134 duplicate studies, 537 articles were screened for title and abstract, of which 80 were reviewed in full. Forty-one studies were included in the review (Table 1). Among the 41 included papers were: Twenty opinion/perspective/communication/special/commentary/correspondence/experience/document reports, eight of reviews, four guidelines, three position papers, two editorials, and four randomized controlled trials (RCT). It is important to emphasize that this study was not conducted on a specific group of people regarding the broad effect of physical activity on COVID-19.

The findings of this review are divided into the following three main themes: (Section 3.1) PA and COVID-19 prevention; (Section 3.2) Impact of PA on patients with acute COVID-19; (Section 3.3) Impact of PA on people without SARS-CoV-2 infection during the pandemic.

### 3.1. PA and COVID-19 Prevention

Keeping a one-meter social distance, wearing masks, washing hands, and vaccination are the effective ways widely used to prevent SARS-CoV-2 infection [51]. PA is also an effective preventive measure during the COVID-19 pandemic. A study revealed that regular exercise could enhance immunity and reduce the incidence of upper respiratory tract infections (URTI) [52]. Klentrou et al., found that regular and moderate PA increased the immunoglobulin (Ig) A concentration, which may be the reason for reducing the incidence of URTI [19]. Several studies summarized that PA could bring various positive effects on the human immune system [53,54,55,56], such as the increase in the blood counts of natural killer (NK) cells, neutrophils, lymphocytes, monocytes, plasma interleukin-6 levels and the function of NK cells, and decrease in inflammation. These help to prevent SARS-CoV-2 infection and severe symptoms. In addition, some studies indicated that PA might enhance the immune response to the influenza vaccine [57,58]. Furthermore, a similar phenomenon was observed in the SARS-CoV-2 vaccine [20], which may enhance the immune protective effect provided by the vaccine. However, Kakanis et al., found that high-intensity PA may cause immunosuppression, which enhances the susceptibility to upper respiratory tract diseases [59]. Campbell and Turner’s “J curve” theory explains that excessive long-term training will suppress immune function, while regular moderate-intensity exercise improves immune function [60]. In conclusion, during the COVID-19 pandemic, regular low/moderate intensity PA is recommended to enhance immunity. At least 150 min of moderate-intensity PA per week are recommended for adults [21], but long-term high-intensity PA should be avoided.

Cardiorespiratory fitness (CRF) reveals the integrated function of various systems and total body health. Studies have revealed that better CRF is associated with lower all-cause mortality [61,62]. Brawner et al., found that CRF may have a negative correlation with the likelihood of COVID-19 hospitalization. And the hospitalization likelihood of men is higher than that of women with similar levels of CRF [45], which suggests that CRF affects the prognosis of patients with COVID-19 and has different effects on patients of different genders. Therefore, improving CRF through regular exercise is helpful in preventing COVID-19. Respiratory muscle training is an effective intervention to reduce pulmonary complications [22]. During respiratory muscle training, repeated compression by inhalation and exhalation inhibits inflammation, such as monocyte chemoattractant protein-1 (MCP-1) production, macrophage infiltration, and TNF-α production [47,63]. More importantly, respiratory muscle training improves the strength and endurance of respiratory muscles and enhances respiratory function [64], which is beneficial to both healthy people and patients with COVID-19. Khoramipour et al., suggested 50–100 times resistance breathing training five days per week for healthy and asymptomatic patients [23].

Mitochondria play a key role in the process of energy production. It utilizes fats, sugars, and proteins by oxidation to generate adenosine triphosphate (ATP) and maintains the life of cells and the normal operation of various functional activities [65], such as preventing endogenous oxidative stress [66]. Active exercise can maintain the activity of mitochondria [66], thereby obtaining more energy, slowing down age-related muscle loss, improving physical function, and preventing related diseases [67,68]. Therefore, enhancing mitochondrial function through PA also contributes to the prevention of COVID-19. Traditional PA combined with resistance training [69] or high-intensity intermittent training (HIIT) [14] has greater benefits for the renewal and function improvement of mitochondria.

In addition, patients with elder age, obesity, cardiovascular disease, liver disease, diabetes, or cancer are high-risk groups for COVID-19. These patients are more likely to develop severe symptoms and even die after SARS-CoV-2 infection [24,51,70]. PA helps prevent and treat obesity, cardiovascular disease, diabetes, liver disease, cancer, and other diseases [10,11,12,25], thus, indirectly reducing the threat of COVID-19. Therefore, PA may bring greater health benefits to these special populations (Figure 2).

### 3.2. Impact of PA on Patients with Acute COVID-19

Respiratory symptoms are the most common symptoms in patients with COVID-19, and if not properly treated, patients may develop into severe cases or even die [2,3]. Several studies indicated that PA might relieve respiratory symptoms in patients with COVID-19 [14,17]. Modified rehabilitation exercise (MRE) is a full-body exercise designed to reduce total airway resistance, smooth airflow, and improve O2/CO2 exchange efficiency. Zha et al., found that after one month of MRE from the Chinese martial art Baduanjin, the prevalence of symptoms such as dry cough, productive cough, difficulty in expectoration, and dyspnea, decreased significantly in hospitalized patients with mild COVID-19 [26]. Li et al., used physical therapist interventions, including body positioning, airway clearance techniques, oscillatory positive end-expiratory pressure, inspiratory muscle training, and mobility exercises, in 16 patients with COVID-19 in the intensive care unit (ICU). The results suggest that physical therapist interventions are useful in improving both respiratory and physical function in ICU patients with COVID-19 [27]. However, at discharge from ICU, peak expiratory flow rate (PEFR) and maximum inspiratory pressure (MIP) were still lower than 80% of the predicted values in some patients. Meanwhile, 46% of the patients had De Morton Mobility Index values below the normative value. This suggests that even after discharge, some ICU patients still need long-term rehabilitation training. A systematic review revealed that early progressive mobilization in ICU patients is feasible and safe to improve function and shorten ICU and hospital stays [71]. A randomized controlled trial (RCT) showed that early PA contributed to the shortening duration of delirium and mechanical ventilation in ICU patients [72]. In addition, vibratory and electromyostimulation may produce a similar effect to traditional PA [73,74,75]. The World Association of Vibration Exercise Experts (WAVEX) recommended whole-body vibration (WBV) exercise as a safe and effective intervention for improving physical function and quality of life (QoL) in hospitalized patients with COVID-19 and also shortening the length of ICU stay [76].

PA can also alleviate the negative psychological impact of COVID-19 on patients. The study by Zhang et al., showed that anxiety and depression levels varied among different COVID-19 patients in the square cabin hospital, with female and lower education levels being associated with higher levels of anxiety and depression. Through the “Baduanjin” exercise, patients’ anxiety and depression were significantly relieved [77]. A systematic review of 23 studies also indicated that PA improves both physical and psychological outcomes in COVID-19 patients [78].

Most guidelines recommend personalized PA for patients with different disease severity [40,41,42,43]. Passive mobilization and posture changes are recommended for sedated or unconscious patients [43,44]. Once the patient’s sedation has diminished, gradual PA should be initiated to avoid physical disability and myopathy. However, patients in the acute phase of COVID-19, especially those with severe symptoms, should not exercise excessively so as not to increase the incidence of respiratory distress and other symptoms [44]. Therefore, in general, proper PA is safe and beneficial for patients with COVID-19.

### 3.3. Impact of PA on People without SARS-CoV-2 Infection during Pandemic

Other than COVID-19 patients, the COVID-19 pandemic also had a negative influence on the physical and mental health of other people.

Studies demonstrated that the COVID-19 pandemic has also caused widespread negative psychological effects on people, such as post-traumatic stress symptoms (PTSS), confusion, anger, etc. [50]. In addition, compared to adult men, adult women are more likely to experience anxiety during the COVID-19 pandemic [79]. Isolation during the COVID-19 pandemic may also lead to unhealthy eating habits, such as overeating and snacking, which may lead to weight gain and nutritional imbalances that impair immune function [38]. PA can improve the psychological state of healthy people and patients, which is beneficial to the prevention and treatment of various diseases [12]. Similarly, Li et al., found that the “Baduanjin” exercise alleviated the anxiety related to COVID-19 and improved the psychological well-being of college students [34]. “Green exercise” refers to physical activity in the presence of nature, which positively impacts mental health. A large-sample study demonstrated that green exercise might improve mood and reduce mental health symptoms during the COVID-19 pandemic [35]. Therefore, PA can effectively improve the mental health of healthy people and patients during the pandemic, offset the negative effects of unhealthy eating habits, enhance immunity, and improve QoL [39].

## 4. Discussion

In summary, different exercise regimens such as respiratory muscle training, lung group training, strength training, and endurance training have been proposed in relevant studies. However, a comprehensive approach should be considered depending on the period of COVID-19 disease and symptoms. The effect of an exercise intervention on chronic disease is different from that of neocon disease, so the way of exercise intervention is also different. It has been suggested that high-intensity interval training is more beneficial than moderate intensity in improving cardiopulmonary function in cardiovascular patients by increasing the body’s demand for oxygen through explosive, rapid, and adequate exercise in a short period of time, causing hypoxia and increasing the patient’s cardiac pumping function. This training modality is not recommended, especially for patients without exercise habits and special populations. In contrast, whole-body exercises in the form of modified rehabilitation exercise (MRE) and full-body vibration (WBV) can be used to improve lung function more safely. Therefore, we recommend that the COVID-19 training program should be planned in stages and that the intensity of physical activity should be gradually increased in a customized health management plan to ensure more comprehensive management of these patients.

Therefore, we believe that different recommendations should be provided for different types of COVID-19 patients. The early stage of acute COVID-19 disease, i.e., 48 h after admission, is not recommended for direct progressive exercise. Reasonable control of exercise intensity through professional guidance is needed in the early resumption of exercise to avoid secondary injury from excessive exercise. Second, exercise in these patients should be measured more objectively to ensure appropriate training. In addition, early patients with COVID-19 have impaired respiratory muscle strength and need some observation time. When the risk period is passed, appropriate interventions in exercise therapy, primarily rehabilitation of the respiratory muscles, can be made. Early rehabilitation programs add mainly respiratory muscle training to maximize the benefit for COVID-19 patients by increasing their respiratory muscle strength and also by adding some resistance training that may help to reduce the problems of COVID-19 sequelae. Regular exercise of moderate intensity is recommended for COVID-19 patients with milder symptoms; moderate aerobic or resistance exercise may be chosen for milder patients; on the one hand, exercise is needed to reduce intracellular and extracellular oxidative stress. On the other hand, it is to improve drug side effects and promote the digestive system to maintain intestinal flora homeostasis. It is recommended to add yoga and tai chi, which are non-pit-like exercise programs, to the exercise rehabilitation program to improve the disease effectively; for patients with severe COVID-19, it is necessary to provide transfer Due to the different comorbidities and clinical manifestations of different patients, and there are also differences between gender and age, it is more recommended to personalize the training program in the development of the training program. Progressive exercise programs, early activity, and multi-component interventions to improve activities of daily living can be provided in ICU units for severe respiratory disease since many patients with severe COVID-19 progress to mechanical ventilation and remain in the hospital for long periods. Effective improvement can be promoted by early exercise, which can help patients recover and improve their confidence through individualized exercise interventions.

As the number of people cured of COVID-19 increases, a large number of people will enter the recovery phase. Patients with COVID-19 experience respiratory distress, pulmonary impairment, and circulatory limitations, including muscle weakness, motor deficits, and decreased exercise capacity during the recovery period, and many studies have shown that unsupervised home training can be conducted using a tele-rehabilitation program (TERECO) [80], conducted via smartphone and monitored by heart rate telemetry. However, we believe that while TERECO has improved physical recovery, it does not allow for timely observation of the patient’s quality of life and psychological status. More in-depth communication is needed at the mental health level. Physician-assisted training is more beneficial to patients’ recovery than unsupervised rehabilitation [81]. Therefore, it is recommended that the patient’s recovery and mental status be monitored through regular follow-up and patient review.

In the prevention phase of neocon disease, the main focus is on improving immune system function, which is more suitable for moderate intensity, while avoiding prolonged high-intensity exercise. 1. Mitochondrial health is one of the factors in the prevention of COVID-19 pathogenesis and endurance training combined with resistance training has a positive impact on mitochondrial production and optimization. In addition, new training methods based on low volume and high intensity, such as high-intensity interval training (HIIT), are suitable for healthy populations and have shown great efficacy in generating new mitochondria and optimizing their function; 2. For the elderly population, multicomponent physical activity should be performed, the exercise load is not easily excessive, and the main exercise components include balance training as well as aerobic and muscle strengthening activities. Pregnant and postpartum women should engage in at least 150 min of moderate-intensity aerobic exercise per week. For special populations, 150 min of moderate-intensity or 75 min of vigorous-intensity physical activity over 3 to 5 days is sufficient to improve physical and mental health and reduce the prevalence of COVID-19 symptoms. Adolescents or preschoolers should engage in 60 to 180 min of moderate-intensity physical activity. Healthy young people should exercise 2–3 times per week for 150 to 300 min per week at moderate or 75 min of high intensity. This will not only improve fitness but also prevent neoconiosis. 3. COVID-19 Regular exercise during confinement can reduce physical damage from sedentary behavior and improve psychological problems. For non-infected people, it is appropriate to increase the time of outdoor activities. The psychological benefits of “green exercise” are greater than those of indoor exercise. During the COVID-19 lockdown, for non-infected people, psychological stress increases over time, which can lead to mental illness. More than 30 min of “green exercise” per day can meet people’s expectations and can effectively combat anxiety and depression and relieve anger, depression, and fatigue. Secondly, experiencing nature is often considered the second most important physical activity motivation when tired of “home exercise”; a short outdoor activity will reduce stress and relieve emotions. Therefore, it is recommended that “green exercise” be maintained three times a week to relieve mental stress. It is also important to avoid the risk of infection from crowds of people.

Overall, both rehabilitation and prevention for COVID-19 patients should be based on a low-intensity exercise in the early stages and a gradual increase in exercise load to reduce the likelihood of musculoskeletal injury. Newly crowned patients should exercise under the guidance of a physician to meet individualized needs and maximize compliance.

## 5. Conclusions

Overall, PA benefits immune function, cardiorespiratory fitness, and mitochondrial renewal and activity, thereby preventing SARS-CoV-2 infection and severe symptoms of COVID-19. Exercise therapy can improve clinical outcomes in patients with COVID-19, such as improved respiratory and motor function and shorter hospital stays. PA also improves the mental health of healthy people and patients, alleviates post-acute COVID-19 syndromes, and improves QoL. COVID-19 affects patients with different characteristics (e.g., male and female) differently [4,5,36,51]. The appropriate PA intensity and types are dissimilar in populations with different characteristics (e.g., children, adult men/women, pregnant/postpartum women) or patients at different COVID-19 stages [12,13,17], and the effects of PA on these populations are also different [29]. Therefore, a safe and effective personalized exercise regimen should be adopted for each individual.

Although current studies have explored various aspects of the impact of PA on COVID-19, there are still some limitations. First, most of the current studies are low-level evidence, and more RCTs with large samples are needed. Second, the intensity and types of PA, which are appropriate for patients at different stages of COVID-19, especially those with elder age and underlying diseases, are quite different from those for healthy and asymptomatic patients. However, comprehensive and specific suggestions and guidelines on PA for these populations are still lacking. Therefore, more high-quality studies are needed to address these questions to provide higher-level evidence for the formulation and optimization of PA strategies for relevant populations.

In conclusion, healthy individuals or patients at different stages of COVID-19 can get both physical and psychological benefits by taking appropriate and personalized PA. Medical professionals, institutions, and the government should facilitate the public’s awareness of PA, organize sports events, implement relevant policies and programs, and build sports facilities to increase people’s PA and improve their health.

## Figures and Tables

**Figure 1 ijerph-19-14108-f001:**
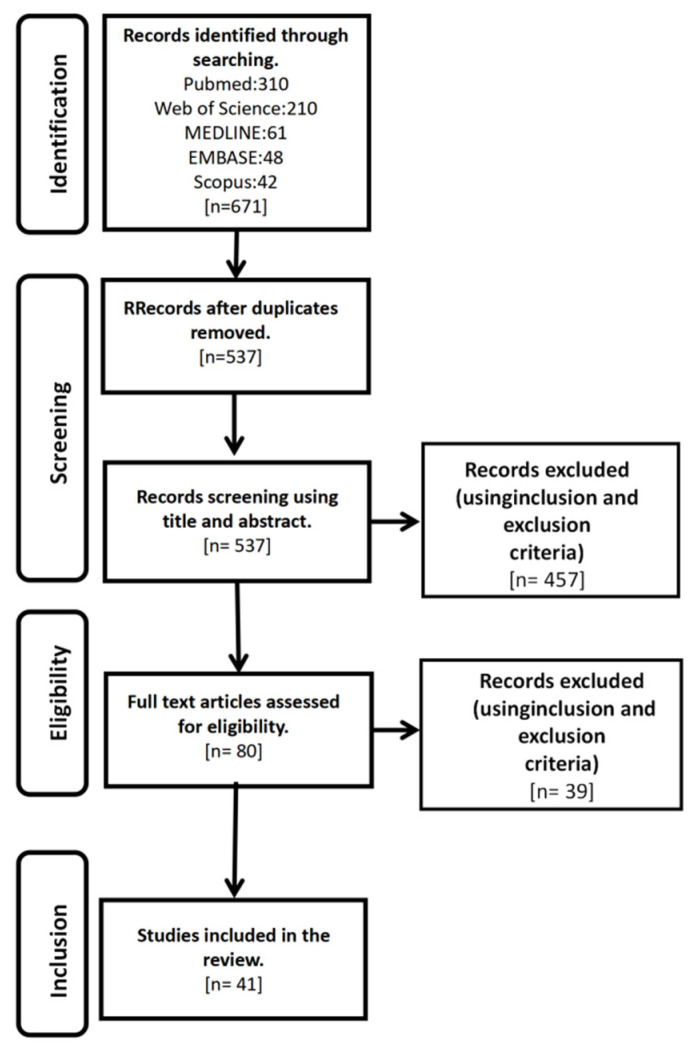
Flowchart detailing the systematic search, screening, eligibility, and inclusion procedure.

**Figure 2 ijerph-19-14108-f002:**
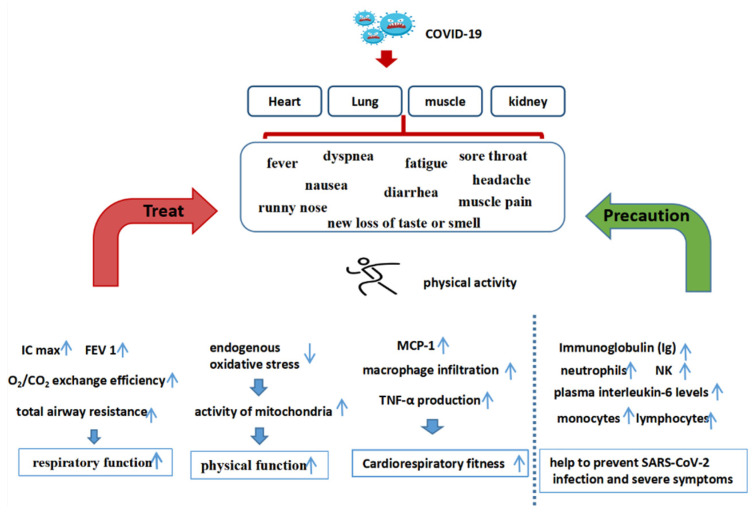
The impact of sports before and after COVID-19.

**Table 1 ijerph-19-14108-t001:** Characteristics of included studies (*n* = 41) and information about localization of papers findings in this narrative review.

Author	Year	Country	Study Design
Bin Zhou et al. [3]	2020	Japan	recommendations
Jing Sha et al. [4]	2021	China	Perspective
Amaya Jimeno-Almazán [16]	2021	Spain	Perspective
Panagiota Klentrou et al. [19]	2002	Canada	Perspective
Bruno Gualano et al. [20]	2020	Brazil	Special
Robert Sallis et al. [21]	2021	United States	Original research
Mei-Yun Liaw et al. [22]	2020	China	Comment
Kayvan Khairpur et al. [23]	2020	Iran	Perspective
Sara Y Tartof et al. [24]	2020	United States	Document reports
Philip O’Gorman et al. [25]	2021	Ireland	Communication
Lulu Zha et al. [26]	2020	China	Opinion
Lei Li et al. [27]	2021	China	Original research
Udina et al. [28]	2021	Spain	Controlled study
Linda Rausch et al. [29]	2022	Austria	Perspective
Monira I. Aldhabi et al. [30]	2022	Saudi Arabia	Experience
Fraser M. Kennedy et al. [31]	2020	England	Perspective
Paola Gonzalo-Encabo et al. [32]	2022	United States	Opinion
Robert Simpson et al. [33]	2020	Canada	Perspective
Keqiang Li et al. [34]	2021	China	Observational Study
Das et al. [35]	2022	United States	Experience
Biolè et al. [5]	2021	Spain	Retrospective Study
Ani Nalbandian et al. [9]	2021	United States	Reviews
Vicente Javier Clemente-Suárez [14]	2020	Spain	Reviews
Veronica Lourenço Wittmer [17]	2021	Brazil	Reviews
Safiya Richardson et al. [36]	2020	China	Reviews
Cox NS et al. [37]	2021	United States	Reviews
Narimen Yousfi et al. [38]	2020	Tunisia	Reviews
Matheus Pelinski da Silveira et al. [39]	2020	Brazil	Reviews
Katrina L. Piercy et al. [12]	2020	United States	Physical Activity Guidelines
Bull et al. [13]	2020	United States	Guidelines
Yuetong Zhu et al. [40]	2020	Japan	Guidelines
Hong-Mei Zhao et al. [41]	2020	China	Clinical Guideline
Robert M. Barker-Davies et al. [42]	2020	England	Consensus Statement
Michele Vitacca et al. [43]	2020	Italy	The Italian Position Paper
Marta Lazzeri et al. [44]	2020	Italy	Position
Clinton A. Brawner et al. [45]	2021	United States	Editorial
Luis Suso-Martí et al. [46]	2021	Spain	Editorial
Katsuhiko Suzuki et al. [47]	2019	Japan	Randomized trial
Kai Liu et al. [48]	2020	China	Randomized
Jian’an Li et al. [49]	2021	China	Randomized
Samantha K. Brooks et al. [50]	2020	England	Controlled trial

## Data Availability

Not applicable.

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
