# Peer review of "Impact of Physical Activity on COVID-19"

_ijerph, 2022, doi:10.3390/ijerph192114108_

Round 1

Reviewer 1 Report

Although it is an interesting study, there many topics need improvement

1. Abstract should have the form "introduction-methods-results-conclusions"

2. Authors mention that they gathered previous studies reffering to physical activity and covid. They should add a flow chart (diagramm), how they did literature search and inclusion exclusion criiteria

3. I would expect in the discussion section to compare the results of the previous studies and not only mention the results. Maybe they could compare the benefits of physical activity in covid patients with other similar conditions

4. Finally this subject is already described extended in previous studies. In my opinion this has not any new information

Author Response

Response to Reviewer 1 Comments

Point 1: Abstract should have the form "introduction-methods-results-conclusions"

Response: Thanks and sorry for the format issue. We have reformatted the abstract which is highlighted in yellow.

Point 2: Authors mention that they gathered previous studies referring to physical activity and covid. They should add a flow chart (diagram), how they did literature search and inclusion exclusion criteria.

Response: Thanks for the question! To answer this question, we have added the "Materials and methods" section to describe our search strategy and eligibility criteria in detail with necessary figures and tables. The literatures included in the study are listed in Table 1 and the literature selection process is shown in Figure 1.These revisions are highlighted in red in the revised manuscript.

Point 3: I would expect in the discussion section to compare the results of the previous studies and not only mention the results. Maybe they could compare the benefits of physical activity in covid patients with other similar conditions.

Response: Thanks for the suggestion! In the revision, we have added the "Discussion" section and discussed how COVID-19 differs from other diseases regarding physical activity. We have also given a scientific approach to physical activity for COVID-19 patients. These revisions are highlighted in yellow on pages 7-8.

Point 4: Finally this subject is already described extended in previous studies. In my opinion this has not any new information.

Response: Thanks for the comments. To provide more new information, we have described our own opinions about the criteria of physical activity for different types of COVID-19 patients, e.g. the early stages of acute illness, mild illness, severe illness, recovery, and non-infected populations. These revisions are included in the "Discussion" section on pages 7-8 in the revised manuscript.

Reviewer 2 Report

Dear authors,

Thank you for examining the effects of physical activity on COVID 19 in your research. However, the research was more like an editorial rather than a review format. Therefore, I could not evaluate your research as a review. How you provide the proofs you put forward in the research, which sources and how you filter them are very important. That's why I find your research editorially acceptable. However, I believe that the research cannot be evaluated as a review as it is. If you are going to present your research as a review, I would definitely recommend you to redesign it by making a good method design (if necessary, a good diagram) in review format.

Your research is suitable for editorial publication as it is, if the editors deem it appropriate. However, I cannot accept your research as a review in the current format.

With my most sincere regards.

Author Response

Response to Reviewer 2 Comments

Point 2: Thank you for examining the effects of physical activity on COVID 19 in your research. However, the research was more like an editorial rather than a review format. Therefore, I could not evaluate your research as a review. How you provide the proofs you put forward in the research, which sources and how you filter them are very important. That's why I find your research editorially acceptable. However, I believe that the research cannot be evaluated as a review as it is. If you are going to present your research as a review, I would definitely recommend you to redesign it by making a good method design (if necessary, a good diagram) in review format.

Your research is suitable for editorial publication as it is, if the editors deem it appropriate. However, I cannot accept your research as a review in the current format.

With my most sincere regards.

Response: Thank you so much for the suggestions and comments. According to the review format, we have revised the manuscript as follows. (1) We have reorganized the manuscript with "2. Materials and methods", "3. Results" and "4. Discussion" sections to make the article more logical and readable. (2) The sources and selection criteria of the articles are described in the "2. Materials and methods" section. (3) A new Figure 1 and a new Table 1 have been added to show the literatures included in the study and the literature selection process, respectively.

Reviewer 3 Report

This article reinforces what we already know:  exercise is good for preventing and recovering from health issues.  However, did you find anything interesting or surprising?  I liked the section that dealt with mental health and green exercise and wonder if these sections could be better developed.

Author Response

Response to Reviewer 3 Comments

Point 1:  This article reinforces what we already know: exercise is good for preventing and recovering from health issues. However, did you find anything interesting or surprising? I liked the section that dealt with mental health and green exercise and wonder if these sections could be better developed.

Response: Thank you very much for your review and comments on the article. In the revision, we have added more detailed discussion in the "4 Discussion" section. The existing studies have described how physical activity should be used to effectively prevent and treat COVID-19, but have not provided specific exercise recommendations for different populations. With literature search and analysis, we come up with the criteria of physical activity for different types of COVID-19 patients, e.g. the early stages of acute illness, mild illness, severe illness, recovery, and non-infected populations.

As for "mental health and green exercise", we have added related content to the discussion on page 9 of the revised manuscript, with the additions marked in green.

Round 2

Reviewer 1 Report

Authors made a great effort and their manuscript is improved. Only a minor correction. In the abstract they should also add in methods the period of literature search

Author Response

Thank you very much for the comment and correction! We sincerely appreciate your consideration. According to the comment, we have added the information of the period of literature search in the abstract. 

Reviewer 2 Report

Thank you so much for your effort. Your manuscript is ready to publish.

Best.

Author Response

Thank you very much for your careful reading and helpful instruction! We sincerely appreciate your time.